# *Apilactobacillus kunkeei* Alleviated Toxicity of Acetamiprid in Honeybee

**DOI:** 10.3390/insects13121167

**Published:** 2022-12-16

**Authors:** Peng Liu, Jingheng Niu, Yejia Zhu, Zhuang Li, Liang Ye, Haiqun Cao, Tengfei Shi, Linsheng Yu

**Affiliations:** 1School of Plant Protection, Apiculture Research Institute, Anhui Agricultural University, Hefei 230031, China; 2Anhui Province Key Laboratory of Crop Integrated Pest Management, Hefei 230031, China; 3School of Plant Protection, Biotechnology Center of Anhui Agriculture University, Hefei 230031, China

**Keywords:** probiotics, insecticide, symbiotic microbiota, survival, 16S rRNA

## Abstract

**Simple Summary:**

The honeybee is an important pollinator and is key in maintaining ecological balance. Insecticides, especially neonicotinoids, are considered as critical factors in colony collapse disorder. However, the question of how to reduce the toxic effect of pesticides on bees has not been comprehensively answered. Probiotics are an important and valid way to combat stress and have the benefit of maintaining healthy honeybees. Our study found that *Apilactobacillus kunkeei*, which was isolated from beebread, can reduce the mortality effect of acetamiprid on honeybees. However, the mechanism is not clear, and we attempted to elaborate on it, based on the symbiotic honeybee microbiota. We found that some opportunistic and pathogenic bacteria invaded the intestinal regions of honeybees under acetamiprid exportation. Meanwhile, the community richness and diversity of symbiotic microbiota were decreased, and the community structure of intestinal bacteria was changed and differentiated. However, with the supplementation of *A. kunkeei*, the community richness and diversity of symbiotic microbiota showed an upward trend, and the community structure was stabilized. These data suggest that *A. kunkeei* may be beneficial to a stabilized community structure which reduces the toxic effects of acetamiprid on honeybees. Our results offer important insights into the application of probiotics and potential probiotics in beekeeping.

**Abstract:**

Nowadays, colony collapse disorder extensively affects honeybees. Insecticides, including acetamiprid, are considered as critical factors. As prevalent probiotics, we speculated that supplementation with lactic acid bacteria (LAB) could alleviate acetamiprid-induced health injuries in honeybees. *Apilactobacillus kunkeei* was isolated from beebread; it significantly increased the survival of honeybees under acetamiprid exportation (from 84% to 92%). Based on 16S rRNA pyrosequencing, information on the intestinal bacteria of honeybees was acquired. The results showed that supplementation with *A. kunkeei* significantly increased survival and decreased pollen consumption by honeybees under acetamiprid exportation. Under acetamiprid exportation, some opportunistic and pathogenic bacteria invaded the intestinal regions. Subsequently, the community richness and diversity of symbiotic microbiota were decreased. The community structure of intestinal bacteria was changed and differentiated. However, with the supplementation of *A. kunkeei*, the community richness and community diversity of symbiotic microbiota showed an upward trend, and the community structure was stabilized. Our results showed that *A. kunkeei* alleviated acetamiprid-induced symbiotic microbiota dysregulation and mortality in honeybees. This demonstrates the importance of symbiotic microbiota in honeybees and supports the application of *Apilactobacillus kunkeei* as probiotics in beekeeping.

## 1. Introduction

As an important kind of pollinator, the western honeybee (*Apis mellifera* L.) plays an essential role in maintaining ecological balance and provides huge commercial value. Meanwhile, honeybees help to chart environmental health maps by collecting pollen, water, nectar, and gum [1]. However, colony collapse disorder (CCD) has destroyed many honeybee colonies in recent years [2]. Honeybees are often exposed to persistent biotic and abiotic stresses that lead to CCD, including environmental pollution, drastic changes in climate, pathogens, parasites, and habitat losses [2,3]. Pesticides are considered as one of the critical contributing factors in CCD. Various types of residual pesticides, such as fungicides, acaricides, insecticides, insect growth regulators, and herbicides, are remained in pollen, beebread, and honey samples [4,5,6].

Neonicotinoids, such as acetamiprid, are some of the most widely applied insecticides in the world. Neonicotinoids are usually applied via foliar sprays, soil drenches, and trunk injections; these lead to high levels of residuals in pollen and nectar. Honeybees are exposed to sub-lethal doses of neonicotinoids and their health is affected. As an integral part of the colony, the quality and quantity of workers directly reflects a colony’s strength. After exposure to neonicotinoids, foragers spend more time grooming and lose flying postural control [7,8]. Meanwhile, neonicotinoids and pathogen synergistic interactions increase individual mortality and negatively decrease lifespan [8,9]. With a decreased number of workers, the colony is less able to collect resources and nurse larvae, which reduces the colony’s capacity to resist extreme environments in winter and summer.

For the improvement of animal health, humans may add probiotics to an animal diet. This also applies to honeybees. Common animal probiotics include lactic acid bacteria (LAB) and Bifidobacterium. LAB are widely distributed in nature and in the digestive tracts of animals. They can increase the nutrition of beebread through fermentation [10] and can increase immunocompetence by stimulating the immune response of the honeybee. Under *Nosema ceranae* infection, significantly increased survival was observed for honeybees after feeding with LAB [11] due to a decreased infection rate [12]. In vivo, it was reported that LAB decreased the mortality and number of honeybee larvae infected by *Paenibacillus larvae* [13,14]. Besides being beneficial to the individual, LAB positively influenced the colony, leading to an increased number of adult honeybees [15]. In the colony, more than 50 species of LAB were isolated from honey, beebread, bee pollen, and honeybees, such as *Apilactobacillus kunkeei*, *Lactobacillus johnsonii*, *Lactobacillus plantarum*, *Lactobacillus apis*, and *Lactobacillus alvei* [12,16,17]. A variety of LAB may be beneficial to maintaining the environmental stability of the hive. Depending on their metabolism (lactic acid, vitamins, short-chain fatty acids, antimicrobial peptide, etc.), *A. kunkeei* plays a crucial role in host health, making it a potential probiotic [18,19]. Gut microbiota imbalance following exposure to the antibiotic oxytetracycline was rescued by LAB supplementation, including *A. kunkeei* [20]. An intestinal microbial community is essential for honeybees. As with other insects, symbiotic microorganisms in honeybees play a critical role in a variety of metabolic and defense functions, including the metabolism of secondary plant metabolites, the modulation of glucose, the regulation of energy, the provision of vitamins, the prevention the infection, and the management of the host immune system and immune responses [21,22,23]. However, some exogenous stresses lead to intestinal microbial disorders and effect the functions of symbiotic microorganisms; these include insecticides and pathogens [24,25]. Exposure to insecticides affects the balance of the intestinal ecosystem, decreases the abundance of core bacteria, and improves the conditions for environmental opportunistic microorganisms [23]. 

In recent years, research on LAB has mainly focused on how to improve growth status, prevent pathogenic infection, regulate the immune system, enhance intestinal metabolism, and maintain a balanced intestinal microbiota and antioxidant mechanism [20,26,27,28]. In insects, LAB have shown potential advantages to sequester and degrade, and even to metabolize, pesticides. They can degrade pesticides and gut cells that are associated with decreased digestive tract absorption. They can also decrease pesticide toxicity and mitigate mortality [29]. In honeybees, LAB are mainly used for their antimicrobial and antipathogenic activities. The present work aims to explore the potential of decreasing acetamiprid toxicity with LAB. The experimental hypothesis is that LAB can decrease acetamiprid toxicity and mortality, as well as attenuating acetamiprid-induced microbiota dysregulation, in honeybees.

## 2. Materials and Methods

### 2.1. Honeybee Rearing

Brood frames were collected from two healthy colonies in the Beekeeping Research Institute, Anhui Agricultural University, Hefei, China. The Varroa destructor population was controlled using Amitraz one month prior to starting the experiment. Late-stage pupae were removed from the colonies and placed into growth chambers at 33 ± 1 °C in darkness and high humidity (relatively humidity 60%), which simulated colony conditions. The honeybees were fostered using a previously described method [30,31].

### 2.2. Culture Conditions of the Bacterial Strain

The LAB strain used in this study was *A. kunkeei* BB1 (GenBank accession number: SUB10515234 Apilactobacillus OM755697), which was isolated from beebread. Unless otherwise stated, the routine culturing of this strain was performed under aerobic conditions at 37 ± 1 °C using a constant temperature shaking table (Tianjin, Honour). The culture medium was MRS normal medium supplemented with 10 g/L D-fructose. The final experimental concentration of Lactobacillus cells was 10^4^ CFU/mL. 

### 2.3. Experimental Design

Acetamiprid (99%, DR) was dissolved in acetone to prepare a stock solution (20,000 mg/L). The acute toxicity of acetamiprid was determined. Based on the results of the pre-experiment, the following concentrations were prepared to determine the LC_50_: 400 mg/L, 300 mg/L, 250 mg/L, 200 mg/L, 150 mg/L, and 0 mg/L. A total of 360 newly emerged honeybees were collected and separated into six groups. They were placed into cages in darkness for four days and fed with freshly prepared mature workers gut homogenate. Then, different concentrations of acetamiprid were added to their diets. The experiment results were recorded at 48 h, and the 48 h LC_50_ was calculated. The procedures and methods for the determination of the oral LC_50_ were based on previous reports [32]. One-tenth of the LC_50_ was selected as the concentration for the following experiment (the results of the acute toxicity of acetamiprid were summarized in the pre- experiment; the LC_50_ value was found to be 259.25 mg/L). Meanwhile, our pre-experiment showed that *A. kunkeei* did not directly depredate acetamiprid in vivo.

Newly emerged honeybees were fed with freshly prepared mature workers gut homogenate in addition to their food (50% sucrose syrup) for four days; the food was replaced every two days [33,34]. This simulated social behavior and ensured that the guts of the newly emerged honeybees established a complete set of symbiotic microorganisms [34,35]. A total of 600 honeybees were collected and separated into 20 cages (into four groups; each group included five experimental replicates). The four groups were designed as shown in Table 1. All honeybees were maintained in an incubator for seven days, and five from each cage were collected on day 11 of the experiment. To study the effects of *A. kunkeei* and acetamiprid exposure (in sucrose and pollen consumption, and in their gut microbiome) on honeybee survival, data were gathered and analyzed for each group.

Acetamiprid was added into sterile sucrose syrup, and *A. kunkeei* was added to sterile pollen.

### 2.4. Nucleic Acid Extraction and Illumina Sequencing

On day 11 of the experiment, five honeybees from 12 cages were sampled for gut microbiota studies (three replicates were randomly selected out of every five experimental replicates). Each honeybee’s whole gut was carefully collected on a clean bench environment and placed in a sterile centrifuge tube (1.5 mL). The gut samples were frozen using liquid nitrogen and stored at −80 °C until use. Microbial community genomic DNA was extracted using the E.Z.N.A.^®^ soil DNA Kit (Omega Bio-tek, Norcross, GA, USA) according to the manufacturer’s instructions. With the primer pair 338F (5’-ACTCCTACGGGAGGCAGCAG-3’) and 806R (5’-GGACTACHVGGGTWTCTAAT-3’), the hypervariable region V3–V4 of the bacterial 16S rRNA gene was amplified. The PCR product was extracted from 2% agarose gel and purified using the AxyPrep DNA Gel Extraction Kit (Axygen Biosciences, Union City, NJ, USA). DNA sequencing was performed on an Illumina NovaSeq PE250 platform (Illumina, San Diego, CA, USA) by Shanghai Major Biotechnology.

### 2.5. Quantitative Polymerase Chain Reaction (qPCR)

Universal 16S rRNA gene primers Eub338 (5- ACTCCTACGGGAGGCAGCAG-3) and 355R (5- GGACTACHVGGGTWTCTAAT -3) were used to amplify the total copies of the16S rRNA genes in the samples with a StepOne Plus Real-Time PCR system (Applied Biosystems, USA). The primers (5- AGCAGTAGGGAATCTTCCA -3) and (5- CACCGCTACACATGGAG -3) were used to amplify the total copies of *A. kunkeei* in the samples. The standard curves of total microbial community and *A. kunkeei* quantification were based on the amplification of the cloned target sequence in the plasmid vector. The quantification of each simple microbial community was estimated based on standard curves and the ChamQ SYBR Color qPCR Master Mix (Vazyme, Nanjing, China), according to the manufacturer’s instructions.

### 2.6. Statistical Analyses

Operational taxonomic units (OTUs) with 97% similarity cut-off were clustered in UPARSE version 7.1, and chimeric sequences were identified and removed [36,37]. The taxonomy of each OTU representative sequence was analyzed using RDP Classifier version 2.2 against the 16S rRNA database (e.g., Silva v138), applying a confidence threshold of 0.7 [38]. All alpha and beta diversities were calculated using Quantitative Insights Into Microbial Ecology (QIIME2) [39], and the graphs of the analysis results were drawn using the R package, GraphPad Prism 9 (GraphPad Inc., La Jolla, CA, USA), and Adobe Illustrator CS6. 

## 3. Results

Under the applied culture conditions, the reduction in survival in the control group (96%) represented the natural deaths of honeybees in a colony without acetamiprid stress or supplementation with probiotic *A. kunkeei*. The treatment experiment (Figure 1a) showed that the survival of the honeybees decreased following acetamiprid exposure (84% compared to the control). Supplementation with probiotic *A. kunkeei* did not lead to an increase or decrease in the mortalities of the honeybees. However, it significantly increased the survival of the honeybees under acetamiprid exportation, from 84% to 92%. To assess the influence of food consumption under acetamiprid exportation and supplementation with *A. kunkeei*, the pollen and sucrose consumption of honeybees were recorded and statistically analyzed (Figure 1b). The results showed that the pollen and sucrose consumption of honeybees were not affected under acetamiprid exportation. Following supplementation with *A. kunkeei*, the sucrose consumption of honeybees did not change, but pollen consumption decreased. The combination with acetamiprid exportation showed similar results.

Compared with the impact of acute oral exposure to acetamiprid supplementation with *A. kunkeei* (Figure 1c), the LC50 value of the AL group (489 mg/L) was higher than that of the A group (250.3 mg/L). This indicated that the toxicity of acetamiprid was reduced in honeybees. To investigate the influence of intestinal bacteria under acetamiprid exportation and supplementation with *A. kunkeei*, four groups and 12 samples were collected. Based on 16S rRNA pyrosequencing, information on the intestinal bacteria of honeybees was acquired. As shown in Figure 2, 374 OTUs were common to all groups. Most OTUs were found in the CK group, while the fewest were found in the A group. Following feeding *A. kunkeei*, the OTUs in the L group were fewer than in the CK group, but there were more in the AL group than in the A group.

As expected, our microbiota composition analysis showed clear differences between the four groups at the genus level (Figure 3). The core genus (*Lactobacillus*, *Snodgrassella*, *Frischella*, *Gilliamella*, *Bartonella*, and *Bifidobacterium*) in honeybee was detected in all samples and its abundance was over 87%, except for in group A, where it was 58%. The abundance of Lactobacillus was the highest in all treatments (i.e., 45%, 30%, 53%, and 47% in the CK, A, L, and AL groups, respectively). Under acetamiprid exportation, some opportunistic bacteria and pathogens invaded the intestinal regions of honeybees, including *Paenibacillus lautus*, *Paenibacillus* sp. The colonization of these bacteria would compress the living area of the core bacteria and reduce the abundance thereof. However, comparing the AL group and the A group, the abundance of core bacteria was higher in the former (94%). The presented work estimated the absolute (via qPCR) abundances of intestinal bacterial species in the gut compartments of honeybees, based on a standard curve. The total 16S rRNA gene and *A. kunkeei* copies of the intestinal bacteria were collected in four groups (Figure 4). The results showed that the total 16S rRNA gene copies of the intestinal bacteria in the AL, A, and L groups were not significantly differentiated from those of the CK group. The copies of *A. kunkeei* in the AL and L groups were significantly increased compared to those of the CK group.

Next, to determine how acetamiprid and *A. kunkeei* influence the abundances of intestinal bacteria on the species level, the presented work screened different species based on the 16S rRNA database and analyzed the difference using Student’s *t*-test (Figure 5). The results showed that the abundances of *Lactobacillus helsingborgensis* (0.25%) and Lactobacillus sp. (0.13%) in the CK group were higher than the A group (Figure 5a). In the L group, the abundance of *Lactobacillus apis* (0.15%) was higher than in the CK group, but the abundance of Lactobacillus sp. (0.02%) was lower than in the CK group (Figure 5b). Comparing the A and AL groups, the different species were *Bifidobacterium* sp. and *Snodgrassella alvi* (Figure 5c). The abundances thereof in the AL group were higher than in the A group.

The biological diversity of intestinal bacteria is an important criterion for assessing intestinal stability and health. The presented work evaluated the influence of intestinal biological diversity under acetamiprid exportation and supplementation with *A. kunkeei* on the α-diversity (Figure 6) and β-diversity (Figure 7). Based on statistical data and software, the presented work obtained some diversity indexes (Student’s *t*-test). The observed species (Sobs) index is regarded as a measurement of community richness in terms of biological diversity. The results of the Sobs index (Figure 6a) showed that the A group had the lowest community richness among all groups. The community richness in the CK and L groups was significantly higher than that of the A group. In the AL group, the community richness was also observed to be higher than in the A group, although this was not statistically significant. The Shannon evenness index (SEI) measures community evenness. The obtained index (Figure 6b) showed that no significant differences existed among groups. The community evenness of the A group fluctuated greatly. The status and queues in the cluster (Qstat) index is usually used to measure community diversity. The obtained Qstat index (Figure 6c) showed that the A group had the lowest community diversity among all groups. The community diversity in the CK and L groups was significantly higher than that of the A group. In the AL group, the community diversity was also observed to be higher than in the A group, although this was not statistically significant. Phylogenetic diversity (PD) was also measured. The results (Figure 6d) showed that the AL group had the lowest community phylogenetic diversity among all of the groups. The community phylogenetic diversity in the CK and L groups was significantly higher than in the AL group. For a more accurate estimation of the biological diversity or the community structure of the intestinal bacteria in all groups, the beta diversity was assessed using principal component analysis (PCA, Figure 7a), principal coordinate analysis (PCoA, Figure 7b), and nonmetric multidimensional scaling (NMDS, Figure 7c). As a common method of dimensionality reduction, PCA visualizes multidimensional data using scatter diagrams. Based on the graphic, we found a variety of intestinal bacteria community structures in the A group. In the AL group, the community structure was clearly stabilized. Similar results were found when complex microbiota information was analyzed using PCoA and NMDS. Overall, compared with the CK group, the community structure of intestinal bacteria had changed, and and the newly differentiated structure had stabilized in AL group compared with the A group. There was little difference between the L groups compared with the CK group regarding the structure of the intestinal bacteria community.

## 4. Discussion

Neonicotinoids and acetamiprid are widely applied to control pests in agricultural and in domestic and public health activities. In agriculture, these compounds have been used to control aphids, hemipterans, lepidopterans, and other pests via foliar sprays, soil drenches, trunk injections, and seed soaks [40,41]. Following its application, residual levels of acetamiprid may occur in pollen, nectar, and water, with sublethal effects, and even mortality, on honeybees. As a widely applied insecticide, acetamiprid is considered to be slightly toxic or practically non-toxic to mammals compared to organophosphorus and carbamate insecticides; however, it has a very distinct toxicity toward pollinating insects (such as honeybees). In our study, we found that sublethal doses of acetamiprid exhibit toxicity on honeybees, significantly decreasing survival rate. This phenomenon was also observed under semi-field conditions [42]. Acetamiprid is a neurotoxin which also reduces sucrose sensitivity, interferes with memory function, and impacts on foraging and learning abilities [43]. Exposure to sublethal levels of acetamiprid also leads to a decrease in the honeybee’s weight and affects its development from larva to adult [44]. Continuous exposure to can lead to death. Therefore, acetamiprid can have adverse effects on honeybee health and colony productivity. In the present study, during exposure to sublethal levels of acetamiprid, sucrose and pollen consumption by honeybees were not significantly different when compared to the control treatments, suggesting that food consumption by honeybees is not influenced by acetamiprid exposure. A previous study showed that sucrose consumption by honeybees decreased significantly following exposure to very high concentrations of thiacloprid, but that it did not change significantly following exposure to low concentrations [45]. We speculated that the characteristics of acetamiprid are similar to those of thiacloprid; both are cyano-substituted neonicotinoids which are moderately toxic to honeybees [43]. Exposure to a high concentration of acetamiprid could induce escape behavior responses of honeybees.

LAB has been widely administered as a probiotic to improve the health of humans and animals. As a common LAB, *A. kunkeei* is widely distributed in the colony environment, including bee pollen, beebread, and combs [46]. Honeybees acquire it through trophallaxis with companions or through the consumption of bee pollen and beebread. *A. kunkeei* was also isolated from the honey crops of nine *Apis* species and three stingless bee species [47]. This suggests that *A. kunkeei* is safe for hosts and is globally present in bees. A previous study showed that the microbiota associated with maternally provisioned host pollen perform critical functions in bee larval nutrition and survival [48]. Meanwhile, metabolic pathways with phenolic acids allow *A. kunkeei* to contribute to the degradation of secondary plant metabolite phenolic acids in pollen, to reduce the toxicity of phenolic acids in pollen [49]. It has both technological and functional beneficial attributes. This suggests that *A. kunkeei* may play an important role in the development and growth of honeybees. It has great probiotic potential. In our research, supplementation with *A. kunkeei* as a potential probiotic in pollen did not induce the escape behavior response, and food consumption did not decrease during the experiment. It also did not increase the mortality of honeybees when compared to the control treatment. Moreover, pollen consumption by honeybees decreased when *A. kunkeei* was added, in comparison with the acetamiprid-exposed and control groups. The reason for this may be that the quality and nutrition of pollen increased after supplementation with *A. kunkeei*, i.e., the litter pollen could provide enough nutrients for development and growth. The decrease in the pollen consumption of the colony following *A. kunkeei* treatment manifested in a significant increase in the quantity of reserve pollen in the colony [50]. As a probiotic, *A. kunkeei* has been applied for the treatment of major honeybee microbial infections, such as *P. larvae* [13], *Melissococcus plutonius* [47], and *N. ceranae* [12]. However, few studies have been conducted to explore the relationship between honeybees, probiotics, and pesticides. Our study shows that supplementation with *A. kunkeei* significantly improved the survival of honeybees following acetamiprid exposure. In order to clarify the mechanism for this, the presented work obtained and analyzed information about the intestinal microorganisms in sample honeybees.

Intestinal microorganisms play an important role in the development of insects based on their secondary metabolites. Intestinal microorganisms positively impact the growth, development, and reproduction of insects via the advanced metabolism of the host [51]. Stable intestinal symbiotic bacteria are beneficial to host health. Meanwhile, intestinal microorganisms enhance the ability of the host to cope with stress from the environment. Our previous research showed that honeybees maintained a stable intestinal bacteria environment during overwintering, and that an abundance of Lactobacillus may help the host to successfully survive the extreme coldness of winter [52]. As an intestinal microorganism, *A. kunkeei* supplementation for honeybees did not have an influence on the diversity and ecological structure of the intestinal bacteria of the host. However, it likely decreased the abundance of Lactobacillus sp. and increased the abundance of *L. apis*. This was similar to previous reported results [50]. This may be contrary to the supplementation with *A. kunkeei*, where sublethal acetamiprid exposure seriously changed the diversity and ecological structure of intestinal bacteria. This was, again, consistent with previous studies [53]. However, the difference in terms of the gut microbiome richness was that the alpha diversity decreased after acetamiprid treatment compared to that of the control. The apparent differences could be partially caused by different doses and lab environments. Other neonicotinoid pesticides, such as thiacloprid [45], thiamethoxam [54], and imidacloprid [54], also cause imbalances in honeybee gut microbiomes. However, there were no significant differences in the diversity and richness of intestinal bacteria after exposure to 1.0 mg/L amitraz [55]. The apparent differences could be partially caused by differences in the pesticide type, i.e., acetamiprid and amitraz. Amitraz has been used globally to control Varroa mites in beekeeping due to its reduced toxicity to honeybees compared to acetamiprid. Some opportunistic and pathogenic bacteria were detected following sublethal acetamiprid exposure. *Odoribacter* has been proven to be an important pathogenic bacteria in humans and cattle [56]. Although the pathogenicity of *Odoribacter* on honeybees has not been determined, it is also a potential pathogen and may impact the health of honeybees. *Paenibacillus* is an important pathogen, causing American foulbrood of honeybee larva; therefore, it is a major hazard to beekeeping. Interestingly, honeybees that were damaged by acetamiprid were shown to significantly improve in health after supplementation with *A. kunkeei*. This suggests that *A. kunkeei* could enhance the pesticide resistance of the host through the maintenance a stable ecosystem, or by increasing the diversity of the intestinal bacteria [57]. As a core genus, the abundance of *Bifidobacterium* was increased; this was beneficial to the host. *Bifidobacterium* was mainly involved in the degradation of the toxic substance trehalose and in the absorption of some carbohydrates [15]. When dealing with the stress of toxicity, the host needs energy and nutrients. The increase of *Bifidobacterium* could effectively enhance the energy intake of the host and assist in efficiently responding to acetamiprid stress. As for other core bacteria, *S. alvi* distributed in the ileum can form an oxygen pressure environment through respiration. The increase in *S. alvi* could inhibit the passage of opportunistic and pathogenic bacteria. This bacterium enhances the stability and persistence of the characteristic honeybee intestinal bacteria. Meanwhile, the redundant fermentation products of Bifidobacterium may be taken up and oxidized by *S. alvi* for energy and carbon [23]. A previous study demonstrated that the gut microbiota may influence detoxification gene expression in the honeybee digestive tract [58]. Overall, *A. kunkeei* can reduce the acetamiprid toxicity of honeybees by increasing microbial diversity and maintaining the ecological structure of intestinal bacteria. These findings were reported in a previous study [59].

## Figures and Tables

**Figure 1 insects-13-01167-f001:**
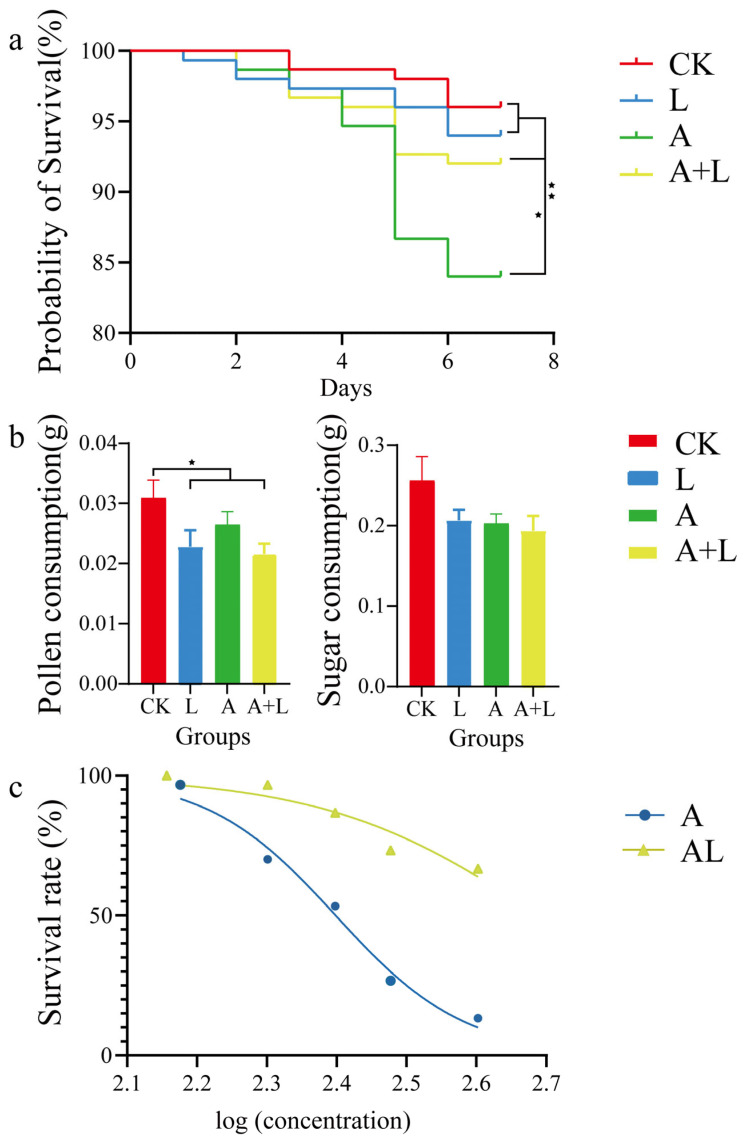
The effect of survival and food consumption under acetamiprid exportation and *A. kunkeei* addition for honeybees. (**a**) The survival of honeybees under acetamiprid exportation and *A. kunkeei* addition. (**b**) Consumption of food for a total of six days under acetamiprid exportation and *A. kunkeei* addition. (**c**) Impact of acute oral exposure to acetamiprid and of *A. kunkeei* addition. ‘*’ represents a statistically significant difference between the two groups at *p* < 0.05. ‘**’ represents a statistically significant difference between two groups at *p* < 0.01.

**Figure 2 insects-13-01167-f002:**
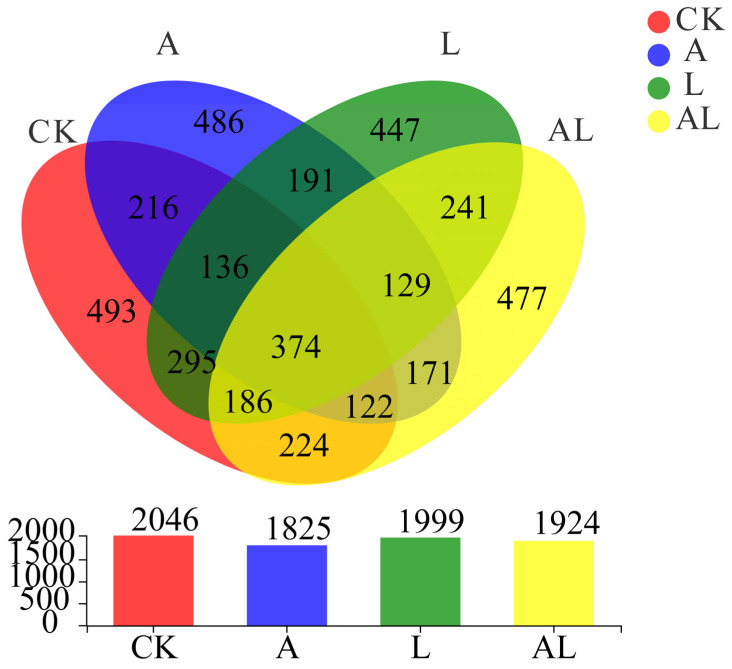
Composition of operational taxonomic units of the intestinal bacteria of honeybees. The number demonstrates the number of OTU, and different colors represent different groups.

**Figure 3 insects-13-01167-f003:**
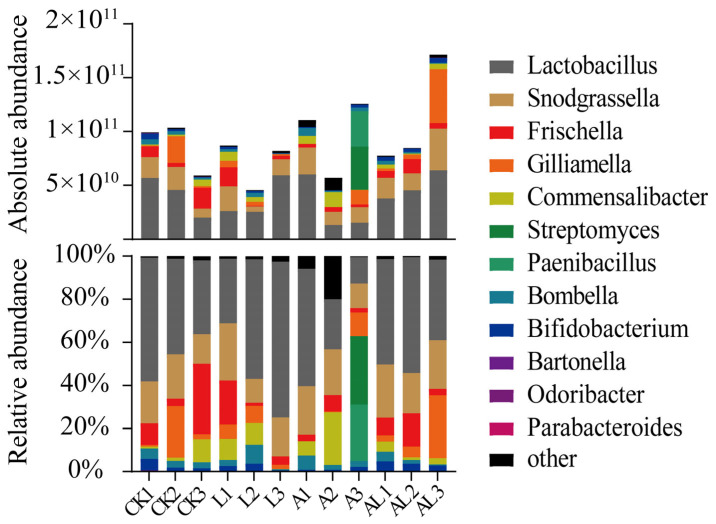
Relative and absolute abundances of the dominant intestinal bacteria of honeybees (genus level). Each bar represents the relative and absolute abundance of each genus in each sample. The transformed absolute abundances were total 16S rRNA gene copies transformed into relative abundance percentages, which provides a crude metric showing how the numbers of each species varied in the four groups.

**Figure 4 insects-13-01167-f004:**
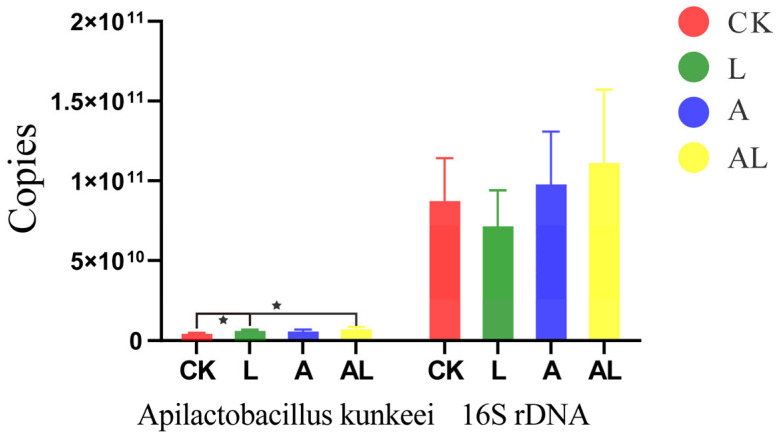
The total 16S gene and *A. kunkeei* copies of intestinal bacteria of honeybees. ‘*’ represents a statistically significant difference between two groups (independent-sample *t* test, *p* < 0.05).

**Figure 5 insects-13-01167-f005:**
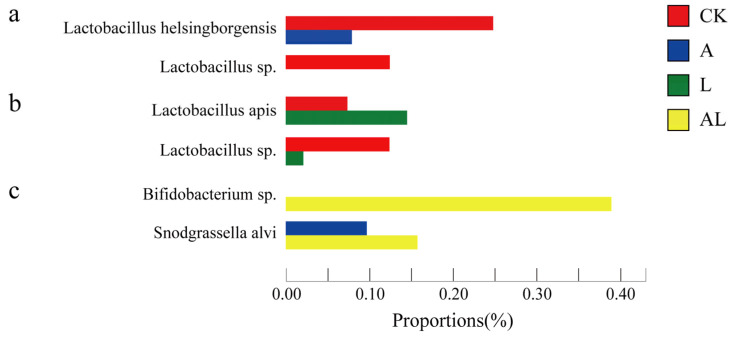
Significant differences in the bacteria in honeybees at the species levels. (**a**) Significant differences between species in the CK and A groups. (**b**) Significant differences between species in the CK and L groups. (**c**) Significant differences between species in the A and AL groups. Each bar represents the relative abundance of each species.

**Figure 6 insects-13-01167-f006:**
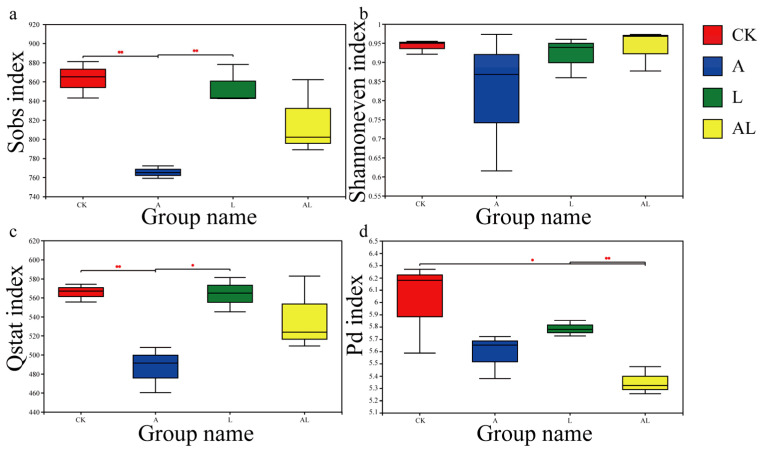
Biological diversity of intestinal bacteria in honeybees at the α level. The level of intestinal bacteria diversity was determined by comparing: the observed species index (**a**); the Shannon evenness index (**b**); the status and queues in the cluster index (**c**); and the phylogenetic diversity index (**d**). ‘*’ represents a statistically significant difference between two groups (independent-sample *t* test, *p* < 0.05). ‘**’ represents a statistically extremely significant difference between two groups (independent-sample *t* test, *p* < 0.01).

**Figure 7 insects-13-01167-f007:**
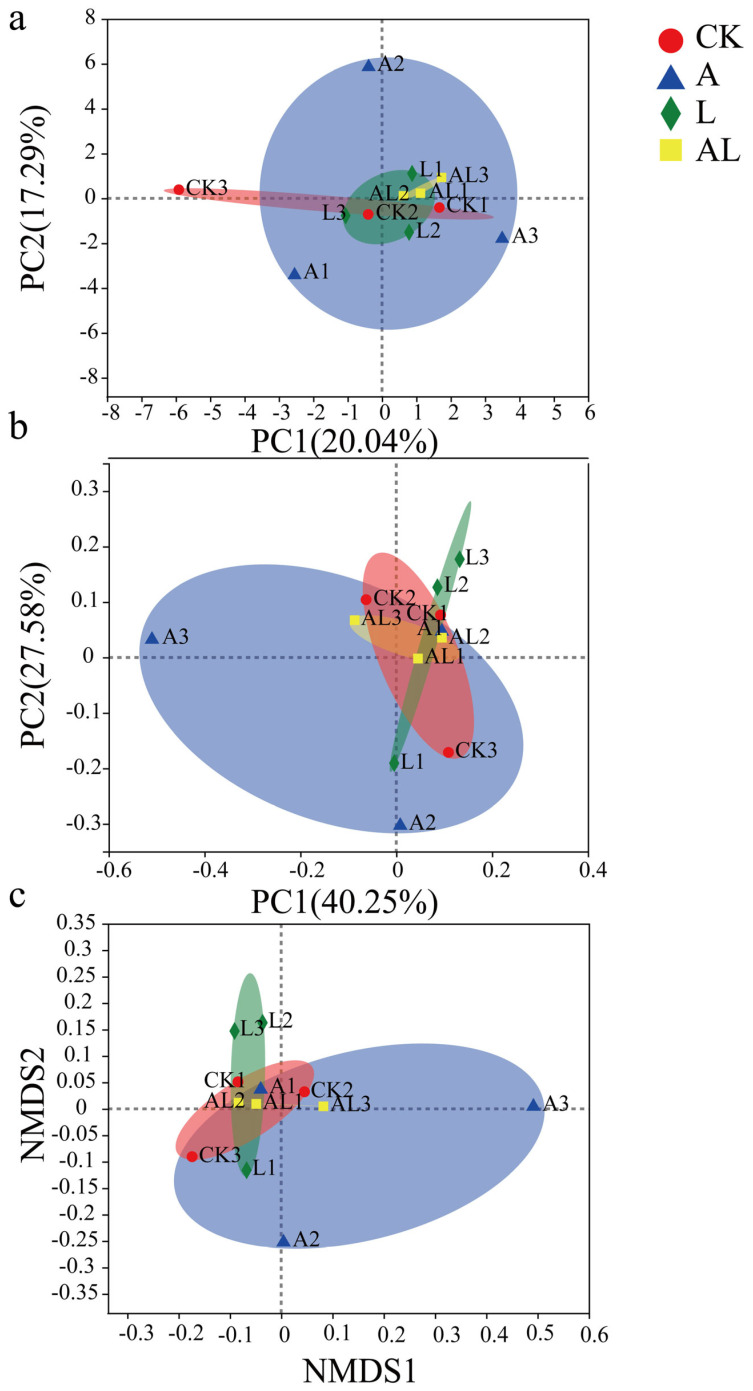
Biological diversity of intestinal bacteria in honeybees at the β level, assessed using: (**a**) principal component analysis; (**b**) principal coordinate analysis; and (**c**) nonmetric multidimensional scaling. Different colors represent different groups. The circle represents confidence ellipse.

**Table 1 insects-13-01167-t001:** Experimental design of the four study groups.

Group	Sterile Sucrose Syrup	Sterile Pollen	Acetamiprid	*A. kunkeei*
CK	+	+		
A	+	+	+	
L	+	+		+
AL	+	+	+	+

## Data Availability

All data generated or analysed during this study are included in this published article. The datasets of microbial community genomic for this study can be found in the SRA database (BioProject ID: PRJNA820132).

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
