# Peer review of "Apilactobacillus kunkeei Alleviated Toxicity of Acetamiprid in Honeybee"

_insects, 2022, doi:10.3390/insects13121167_

Round 1

Reviewer 1 Report

Dear authors,

The manuscript appears to be interesting, but it was difficult for me to read it. It needs major english editing. I tried to address many of the issues, please see the pdf (attached) with the suggestions.

Regards

Author Response

Dear reviewer:

I am very grateful to your comments for the manuscript. According with your advice, we amended the relevant part in manuscript. Some of your questions were answered below.

Thank you very much for your help.

Responds to the comments:

  1. We've changed the “alleviated” of line 15 to “alleviate”, Thank you for your suggests!
  2. Good advice! We added “to” in front of “persistent” in line 35.
  3. Good suggestion! We added “the” in front of “critical” in line 37.
  4. Thank you! We've changed the “to” of line 44 to “of the”.
  5. Good advice! We added “,” in front of “from” in line 61.
  6. Thank you for your suggests, we have added the "et.al" in line 65.
  7. Good suggestion! We've changed the “effected” of line 74 to “effects”.
  8. Thank you! We added “,” in front of “insecticide” in line 75.
  9. We've changed the “balanced” of line 80 to “balance”, Thank you for your suggests!
  10. Thank you! We've changed the “presented” of line 85 to “present”.
  11. Good suggestion! We've changed the “could” of line 86 to “can”.
  12. Good suggestion! We added “the” in front of “following” in line 111.
  13. We've changed the “The” of line 111 to “the”, Thank you for your suggests!
  14. Thank you for your suggests, we have removed the "Meanwhile" from line 127 and changed the “to” of line 127 to “To”.
  15. Good suggestion! We added “in” in front of “sucrose” in line 128.
  16. Thank you for your suggests! We added “in” in front of “their” in line 128.
  17. Thank you! The sentence in line 129 is changed to “it was gathered and analyzed the data for each group”.
  18. Thank you! We've changed the “of” of line 148 to “from”.
  19. Good advice! We've changed the “cutoff” of line 155 to “cut-off”.
  20. Good suggestion! We've changed the “QIIME2” of line 155 to “Quantitative Insights Into Microbial Ecology (QIIME2)”.
  21. Thank you for your suggests, The "It is a known fact that LAB improves the health of honeybees as probiotics. The health of honeybees was affected under insecticide exposure. To evaluate the efficacy of LAB under insecticide exported, the presented work designed a week treatment experiment under laboratory conditions with A. kunkeei and acetamiprid." has been removed from lines 163 to 166.
  22. Good suggestion! We added “In the case of” in front of “Supplement” in line 170.
  23. Good advice! We added “,” in front of “except” in line 192.
  24. Thank you! We've changed the “The” of line 206 to “the”.
  25. Good suggestion! We've changed the “difference species was” of line 212 to “different species were”.
  26. Thank you! We've changed the “was” of line 213 to “were”.
  27. Good advice! We've changed the “supplement” of line 216 to “supplemented”.
  28. Thank you for your suggests, We've changed the “Sobs” of line 218 to “Observed species”.
  29. Thank you for your suggests, We've changed the “Shannoneven” of line 223 to “Shannon evenness index (SEI)”.
  30. Good suggestion! The "Shannoneven" has been removed from line 224.
  31. Thank you for your suggests, We've changed the “Qstat” of line 226 to “The status and queues in the cluster (Qstat)”.
  32. Thank you! We've changed the “product” of line 239 to “the”.
  33. Good advice! We've changed the “the presented work” of line 239 to “we”.
  34. Thank you! We've changed the “that” of line 240 to “a”.
  35. Good suggestion! This sentence on line 241 has been changed to "within AL group community structure comparison is clearly homogeneous."
  36. Thank you for your suggests, Sentence "structure of honeybee intestinal bacterium community in A group was changed and differentiated." is changed to "A group community structure of honeybee intestinal bacterium has changed and differentiated." on lines 243 to 244.
  37. Thank you for your suggests, Sentence "Supplement with A. kunkeei, the difference between AL groups was decreased compared with A group" is changed to "When supplemented with A. kunkeei, the AL groups dimished their difference in comparison with A group" on lines 244 to 245.
  38. Good suggestion! Sentence " and the similar structure of honeybee intestinal bacterium community compared with CK group" is changed to " when compared with CK group, regarding the similar structure of honeybees intestinal bacterium community" on lines 247 to 248.
  39. Good suggestion! The "species between" on lines 280 and 281 has been changed to " between species of".
  40. Thank you! We've changed the “bacteriuml” of line 287 to “bacterium”.
  41. Good advice! We've changed the “would” of line 311 to “can”.
  42. Good suggestion! We've changed the “disappear” of line 312 to “disappearance”.
  43. Thank you! We've changed the “Therefore” of line 312 to “So”.
  44. Good suggestion! The "is" has been removed from line 315.
  45. Good advice! We've changed the “differentiated” of line 315 to “different”.
  46. Thank you! We added “the” in front of “control” in line 315.
  47. Good suggestion! We've changed the “induced” of line 336 to “induce”.
  48. Good advice! We added “,” in front of “and” in line 337.
  49. Good suggestion! We've changed the “was” of line 337 to “has”.
  50. Good advice! We added “the” in front of “experiment” in line 337.
  51. Thank you! We added “when” in front of “added” in line 339.
  52. Good suggestion! The "was" has been removed from line 339.
  53. Good advice! We've changed the “when compare to” of line 340 to “in comparison with”.
  54. Thank you! We've changed the “treatment” of line 340 to “treatments”.
  55. Good suggestion! We've changed the “was” of line 341 to “has”.
  56. Good advice! We've changed the “would” of line 343 to “can”.
  57. Good suggestion! We've changed the “was” of line 343 to “is”.
  58. Thank you! We added “, as” in front of “it” in line 344.
  59. Good suggestion! We've changed the “there were little” of line 346 to “few”.
  60. Good advice! We've changed the “had” of line 347 to “have”.
  61. Thank you! We've changed the “honeybee” of line 350 to “honeybees”.
  62. Thank you! We've changed the “in” of line 350 to “In”.
  63. Good advice! We've changed the “plays” of line 352 to “play”.
  64. Good suggestion! We added “the” in front of “host” in line 355.
  65. Good suggestion! We added “the” in front of “host” in line 356.
  66. Thank you! We've changed the “coped” of line 356 to “cope”.
  67. Good advice! We added “that” in front of “dynamics” in line 358.
  68. Good suggestion! We added “have” in front of “helped” in line 358.
  69. Good suggestion! We added “the” in front of “host” in line 361.
  70. Thank you! We've changed the “increased” of line 362 to “increase”.
  71. Good suggestion! We added “the” in front of “supplement” in line 363.
  72. Thank you! The "was" has been removed from line 375.
  73. Thank you! We've changed the “enhanced” of line 382 to “enhance”.
  74. Good advice! We added “the” in front of “pesticide” in line 382.
  75. Good suggestion! We've changed the “for” of line 364 to “to the”.
  76. Good advice! We added “the” in front of “host” in line 386.
  77. Thank you! We've changed the “respondes” of line 388 to “respond”.
  78. Good suggestion! We've changed the “and” of line 388 to “can”.
  79. Thank you! We've changed the “formed” of line 389 to “formed”.
  80. Good advice! We added “an” in front of “oxygen” in line 389.
  81. Thank you for your suggests, We've changed the “Previous” of line 393 to “A previous”.
  82. Good suggestion! We've changed the “reducing” of line 395 to “can reduce”.
  83. Thank you! We've changed the “increaseing” of line 395 to “increasing”.
  84. Good suggestion! We've changed the “can” of line 395 to “and”.
  85. Thank you for your suggests, Sentence "This was similar to previous speculation" is changed to "Findings were according to a previous study" on lines 396 to 397.
  86. Thank you for your good suggests! The punctuation errors were revised in the article.

Reviewer 2 Report

The work by Peng Liu and colleagues shows potential capabilities of decreasing acetamiprid toxicity with Lactic Acid Bacteria. LAB. decrease acetamiprid toxicity and decrease mortality while may attenuate acetamiprid-induced microbiota dysregulation in honeybees. The results show interesting effects, similar to what has been described in other works using other insecticides. I have found some details in the ms that need to be revised, which hopefully will improve the quality of the present work.

1)              Title: Please change it to something less descriptive.

2)          For all the figures it is necessary to increase the size for a better representation. 3)          Keywords: Please do not include words mentioned in the title of the ms. 4)          In the description of the figures, abbreviations should not be used.

5)              Abstract: you have to give percentages of the main results.

6)              There is no conclusion in the ms.

7)              Line 101:   at 33±1°C in darkness.

8)              L 114-115: Give a brief description of the toxicity tests. Also mention why you chose 150, 200, 250 300 and 400 mg/L, these concentrations are not too high compared to the litterature.

Author Response

Dear reviewer:

I am very grateful to your comments for the manuscript. Thank you very much for your attention and comments on our paper. We have carefully considered your kind advices and detailed suggestion. Based on valuable suggestions, we have revised the manuscript.

Thank you very much for your help.

Responds to the comments:

1 Thank you for your suggests! We've revised the title.

2 Good suggestion! The figures were revised and increased the size in article.

3 Good advice! We've changed the keywords in article.

4 Thank you! All description of the figures were revised in article.

5 Thank you! The abstract was revised and complemented in article.

6 Good suggestion! The conclusion was complemented in abstract.

7 Thank you! The error was revised in article.

8 Good suggestion! The Steps and methods of toxicity tests were descripted in article. These concentrations of acetamiprid were designed based on the results of pre-experiment. The honeybees of pre-experiment were form same colony.